# An Optimized Thermal Feedback Methodology for Accurate Temperature Control and High Amplification Efficiency during Fluorescent qPCR

**DOI:** 10.3390/bioengineering9060237

**Published:** 2022-05-28

**Authors:** Kangning Wang, Yangyang Jiang, Yu Guo, Mingkun Geng, Wenming Wu

**Affiliations:** 1Institute of Biological and Medical Engineering, Guangdong Academy of Sciences, Guangzhou 510000, China; wkn19950104@163.com (K.W.); jiangyy1994@foxmail.com (Y.J.); 2School of Mechanical and Electrical Engineering, Guangdong University of Technology, Guangzhou 510006, China; guoy_21@163.com; 3Changchun Institute of Optics, Fine Mechanics and Physics (CIOMP), Chinese Academy of Sciences, Changchun 130033, China; g215908271@163.com; 4Graduate School, University of Chinese Academy of Sciences (UCAS), Beijing 100049, China

**Keywords:** Conventional PCR, quantitative real-time PCR, temperature feedback system, temperature calibration and optimization, fluorescence detection

## Abstract

Traditional qPCR instrument is combined with CMOS and a personal computer, and a photoelectric feedback automatic fluorescence detection system is designed to realize quantitative real-time PCR. The key to reaction efficiency lies in how to ensure that the temperature of the detection reagent completely matches the set temperature. However, for most traditional real-time fluorescent PCR systems, the temperature cycling is controlled by detecting the temperature of the heating well plate. It cannot directly measure the temperature in the reaction reagent PCR tube, which will cause the deviation in the actual temperature of the reagent to be as expected. Therefore, in this paper, we raise a method of directly detecting the temperature in the reaction tube of the reagent during the temperature cycling is adopted. According to the deviation from the expected value, the set temperature of the PCR instrument is adjusted to make the actual temperature of the reagent closer to the expected value. Through this method, we also realized the temperature calibration and optimization of the TEC circulation system we built. Experiments show that this low-cost, portable real-time quantitative PCR system can detect and analyze pathogens in situ.

## 1. Introduction

PCR is an in vitro nucleic acid amplification technique developed in the mid-1980s. It has many outstanding advantages, such as strong specificity, high sensitivity, high production efficiency, fast, simple, good repeatability, easy automation, and so on [1]. Amplify the gene of interest or the DNA fragment to be studied in a test tube to 100,000 or even 1 million times in a few hours [2]; a sufficient amount of DNA can be amplified from a hair, a drop of blood, or even a cell analysis and test. At present, PCR instruments are widely used in the field of biomedical engineering [3,4]. According to the difference of DNA amplification purpose and detection standard, PCR instruments can be divided into three types: traditional PCR instrument, real-time quantitative PCR instrument, and digital PCR instrument [5,6,7,8,9]. At present, the most widely used are the traditional PCR instrument and the real-time PCR instrument. The traditional PCR instrument only has the function of temperature cycle and can only observe the amplification results with the help of electrophoresis. Based on the traditional PCR instrument, the fluorescence signal acquisition system and computer analysis and processing system are added to become the real-time quantitative PCR instrument, which can judge The concentration of template DNA by measuring the relative fluorescence intensity [10,11]. The fluorescence signal acquisition system collects the fluorescence signal of each temperature cycle in real time and outputs the real-time quantitative results through the computer analysis and processing system [12]. This PCR instrument is called real-time fluorescence quantitative PCR instrument, that is, a qPCR instrument. However, the cost of qPCR is usually about 10 times that of traditional PCR [13].

How to realize a low-cost real-time fluorescence quantitative PCR system is a hot topic in the world [13,14,15,16,17]. For example, the researchers made a simple transformation of the traditional PCR to realize the quantitative PCR, which combines the traditional PCR instruments with smartphones and personal computers and designed an automatic fluorescence detection system with photoelectric feedback to achieve quantitative real-time PCR. The improved PCR instrument mainly includes a thermal cycle system, temperature feedback system, and optical feedback system. Among them, The stability of the temperature feedback system is usually monitored by an infrared (IR) camera [12,18].

The core of PCR is the temperature cycle. The key to the reaction efficiency is to ensure that the actual temperature of the reagent matches the set temperature [19].

However, all commercial PCR instruments and a large number of laboratory-manufactured real-time fluorescent PCR systems are redesigned based on traditional PCR instruments. They all carry out the closed-loop temperature control of the reagent by measuring the temperature of the thermal cycling system, which is to meet the requirements of accurate heat transfer in each reaction well [20]. When there is a large deviation between the measured temperature and the actual temperature of the regent in centrifugal tubes, the temperature feedback is not accurate, and the actual temperature of the reagent deviates greatly from the set temperature, which finally causes that the template DNA may not be able to be amplified normally.

To solve the above problems, a new method for directly detecting the temperature of reaction reagent in the process of temperature cycle is proposed in this paper. The result shows that the thermal cycle temperature feedback system can accurately correct the temperature deviation between the actual temperature and the set temperature of the reagent. On this basis, the temperature of the TEC constructed thermal cycling system is further calibrated and optimized, and a low-cost, portable real-time quantitative PCR system for in situ detection and analysis of pathogens is realized.

## 2. Materials and Methods

### 2.1. Homemade Real-Time PCR System

The real-time PCR system consists of two parts: a thermal cycler and a fluorescence imaging system. The thermal cycler was mainly composed of a thermoelectric cooler (TEC; 25416AC 9Z30N1) and a temperature controller (TCM-M115, Yexian technology, Chengdu, China). A 30-well plate with a PT1000 thermistor was placed on top of the TEC, which is placed on a heat sink with a cooling fan. The PCR system was regulated by a PID temperature controller based on sensor feedback. The fluorescence imaging system consists of a CMOS camera (E3ISPM20000KPA, Kuy Nice, Beijing, China) with a magnifying lens (Chuang Wei, Shenzhen, China), an LED (XPE60W, Cree, Durham, NC, USA) with a dominant wavelength of 470 nm, a condensing lens, and two commercial filters (Xintian Bori, Beijing, China). The 500 nm cut-off emission filter was fixed in front of the CMOS lens, which used for capturing the fluorescence images. The excitation light from the LED, which was filtered by a band-pass excitation filter (470 ± 20 nm).

The real-time PCR system was connected with a computer, which can program the thermal cycler, capture the fluorescence image in each temperature cycle, and process the data to generate the amplification curve.

### 2.2. Commercial PCR Instruments

Two commercial PCR instruments are mentioned in this paper, one is a commercial PCR instrument (LifeECO, Bioer, Hangzhou, Zhejiang, China) which was upgraded with its temperature cycle, and the other is a commercial QPCR instrument (Bio-Rad CFX Connect, Bio-Rad, Hercules, CA, USA) that compares the amplification curve with the homemade PCR instrument.

### 2.3. Temperature Measurement

The temperature was measured in 0.2 mL PCR tubes containing 40 µL of mineral oil (M8410, Sigma-Aldrich, St. Louis, MO, USA). PT1000 thermistors (Hayashi Denko, Toyama-ken, Namerikawa-shi, Nakamura, Japan) and temperature controllers were used to record the data. The temperature of 12 different well positions of commercial PCR thermocycler and homemade PCR thermocycler were measured. The specific locations were shown in Figure 1. The average value of the temperature measured by the 12 channel sensors in the indoor temperature was taken as the reference value, the initial temperature was homogenized, and the subsequently measured temperature was corrected.

### 2.4. Amplification Program

The PCR amplification program was set as follows: the pre-denaturation step at 95 °C for 3 min, the denaturation step at 95 °C for 15 s, and the annealing/extension step at 60 °C for 35 s. After the upgrade, the temperature of the commercial PCR thermocycler was set to 96.8 °C, 97.2 °C, and 60.5 °C, respectively. The temperature of the homemade PCR thermocycler was set to 97.2 °C, 100.4 °C, and 60.6 °C, respectively. The temperature of 2 min in the pre-denaturation step, 10 s in the denaturation step, and 20 s in the annealing/elongation step were selected for statistical analysis.

### 2.5. Reagents

The PCR reagent of the SARS-CoV-2 was composed of 1× Premix Ex Taq HS (TaKaRa Biotechnology (Dalian) Co., Ltd., Dalian, China), 1× EvaGreen Dye (31000, Biotium, CA, USA), 500 nM of forward and reverse primers, and 104 to 106 copies per µL SARS-CoV-2 CDC positive plasmid (Sangon Biotech, Shanghai, China). Each test requires 20 µL of reagent and 15 µL mineral seal above that.

The PCR reagent of the Escherichia Coli was composed of 1× Premix Ex Taq HS, 250 nM of forward and reverse primers, and 1 × 105/mm^3^ of the suspension of Escherichia coli bacteria. Each test requires 20 µL of reagent and 15 µL mineral seal above that. The primer sequences were as Table 1.

Agarose powder (V900510, 2%, Sigma-Aldrich, St. Louis, MO, USA), DL2000 DNA marker (Jialan, Beijing, China), 0.5× TBE buffer (PH1755, Phygene Life Sciences Co., Ltd., Fuzhou, China), and Nucleic Acid GelStain (Gel-Green, KeyGEN Biotechnology, Nanjing, China) were used for agarose gel electrophoresis, which was detected at 302 nm with UV illuminator (ZF1-IIN, JIAPENG Co., Shanghai, China).

## 3. Results

Through the upgraded thermal cycle temperature feedback system, the corresponding temperature is more accurate. The specific operation is to cover the centrifuge tube when detecting the temperature in the centrifuge tube, open a small hole above the centrifuge tube, and place the temperature sensor into the small hole, which can directly measure the temperature in the centrifuge tube and obtain the temperature of the reagent in the tube. The experiments mentioned above are carried out by this method, and the following results are obtained.

### 3.1. Temperature Curve of PCR Amplification Program

#### 3.1.1. Commercial PCR (LifeECO)

The actual temperature curve of the reagent in the centrifuge tube placed in 12 randomly selected wells in Figure 1 during the initial and the upgraded PCR amplification program is shown in Figure 2 and Figure 3. Obviously, before upgraded the temperature setting of the Commercial PCR, the actual temperature of the reagent in the 12 wells is lower than 95 °C in the pre-denaturation step and the denaturation step and less than 60 °C in the annealing/extension step. After adjusting the set temperature of the Commercial PCR instrument, the actual temperature of the reagent in the centrifuge tube in the 12 wells is much closer to the required temperature in the three steps of the temperature cycle.

#### 3.1.2. Homemade PCR

After adjusting the setting temperature of the home-made PCR instrument, the improvement of the accuracy of the actual temperature is similar to that of the commercial PCR.

### 3.2. The Deviation between the Actual Temperature Amplification Program

#### Commercial PCR (LifeECO)

As shown in Figure 4a,c, during the initial temperature cycle procedure, the actual temperature of the reagent in 12 wells in the three steps tends to be on the low side. The maximum deviation of the two steps of Pre-denaturation and Denaturation is close to −3 °C, and the average deviation of each well is concentrated in the range of −1.5 °C~2.5 °C. In the Annealing/Extension stage, the actual temperature deviation of the reagent with 12 wells is relatively small, and the average deviation is concentrated in the range of −0.75 °C~0.25 °C. During the post-upgrade temperature cycle process, the trend of the low actual temperature of the reagent at 12 wells in the three steps was eliminated, as shown in Figure 4b,d. The deviation values in the two steps of Pre-denaturation and Denaturation are greatly reduced, and the average deviation is concentrated in the range of −0.5 °C~0.5 °C. The average deviation of the Annealing/Extension stage is concentrated in the range of −0.25 °C~0.25 °C, and the absolute deviation is reduced.

When the home-made PCR instrument carries out the initial temperature cycle program, similarly, the actual temperature of the reagent with 12 wells in the three steps tends to be on the low side. Among them, the deviation in the Denaturation step is the most serious, and the maximum deviation is close to −9 °C. The average deviation of each well in the Pre-denaturation stage, which is concentrated in the range of −3 °C to −1 °C. In the Annealing/Extension step, the actual temperature deviation of the reagent with 12 wells is relatively small, and the average deviation is concentrated in the range of −1 °C to 0 °C. During the post-upgrade temperature cycle process, the trend of the low actual temperature of the reagent at 12 wells in the three steps was eliminated, as shown in Figure 5b,d. The deviation values of the Denaturation stage decreased greatly, and the average deviation values of 12 well in the Pre-denaturation and Denaturation stages were concentrated in the range of −2 °C to 1 °C. In the Annealing/Extension step, the average deviation of each well is concentrated in the range of −0.5 °C to 0.5 °C. Although the temperature uniformity of each well has not been improved, the absolute deviation has been reduced, so it is closer to the target temperature.

### 3.3. Results of DNA Amplification after Upgrade

We used a home-made PCR instrument with an upgraded temperature cycle (the fluorescence detection system is not used) and a commercial PCR instrument with an upgraded temperature cycle to amplify E. coli DNA. The electropherogram of the amplification products is shown in Figure 6. The ladder is shown on the left, and the two lanes on the right of the ladder represent the products (492 bp) gained by the commercial PCR instrument (LifeECO) and the homemade PCR instrument. After calculation of the gel intensity by ImageJ, the amplification efficiency is about 92.5% of that of the commercial PCR. The intensity and position of the bands in the electropherogram show that both the upgraded commercial PCR instrument and the upgraded self-made PCR instrument can amplify the correct products for Escherichia coli with high efficiency. At the same time, the amplification efficiency of the self-made PCR instrument is very close to that of the commercial PCR instrument.

The relative fluorescence intensity curve shown in Figure 7b is obtained by analyzing the fluorescence intensity of each image obtained by the homemade PCR instrument using ImageJ software. The Ct values of the three curves were 22.28, 25.54 and 28.90, respectively, and the R2 value was 0.99966. The relative fluorescence intensity curve of commercial qPCR is shown in Figure 7a, and the Ct values of the three curves are 22.6, 26.16, and 29.07, respectively. The results show that the amplification curve (Figure 7b) obtained by the homemade PCR instrument after temperature cycle upgrade is similar to that of the commercial qPCR instrument (Figure 7a), which shows that the amplification efficiency of the homemade qPCR instrument is basically the same as that of the commercial qPCR device.

To sum up, our optimization method for the temperature cycle of the commercial PCR instrument and the self-made PCR instrument successfully makes the actual temperature of the reagent in the centrifuge tube closer to the set temperature. However, this method still has many shortcomings, and many aspects have not been discussed in this paper. First of all, this method only reduces the deviation between the actual temperature of the reagent and the set temperature but does not optimize the fluctuation of the actual temperature in the sampling process. And the optimization of the temperature cycle is only reflected in the temperature dimension, and the time of each stage is not optimized. Secondly, the selection of the well position of the PCR instrument is random, and whether there is a relationship between the temperature deviation and the well is not considered. We will also discuss the above shortcomings in the follow-up study.

## 4. Conclusions

This paper proposes a method for directly detecting the temperature of the reaction reagent in the reaction tube during the temperature cycling, which can make the temperature of the detection reagent closer to the set temperature, thereby controlling the temperature cycling process of the PCR instrument more accurately. All experimental data show that by this method, the temperature cycling system of the modified real-time fluorescent PCR instrument is optimized, and a low-cost, portable real-time quantitative PCR system can be used to detect and analyze pathogens in situ.

## Figures and Tables

**Figure 1 bioengineering-09-00237-f001:**
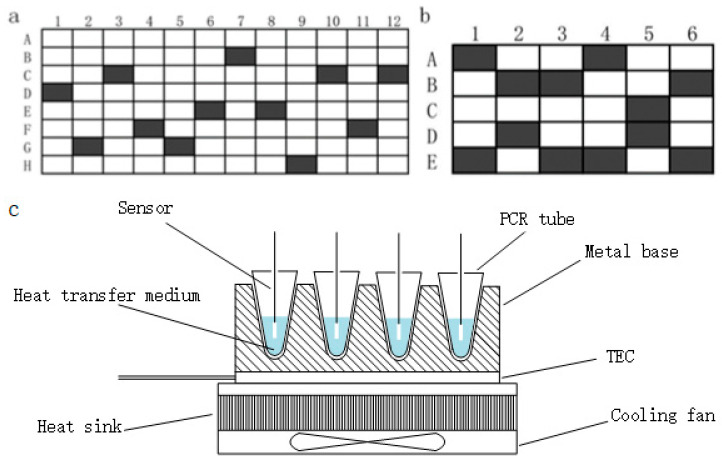
The temperature was measured in 0.2 mL PCR tubes at 12 random well positions. (**a**) The 12 well positions of the commercial PCR thermocycler were D1, G2, C3, F4, G5, E6, B7, E8, H9, C10, F11, and C12. (**b**) The 12 well positions of the homemade PCR thermocycler were A1, E1, B2, D2, B3, E3, A4, E4, C5, D5, B6, and E6. (**c**) Module structure diagram of self-made PCR temperature detection system.

**Figure 2 bioengineering-09-00237-f002:**
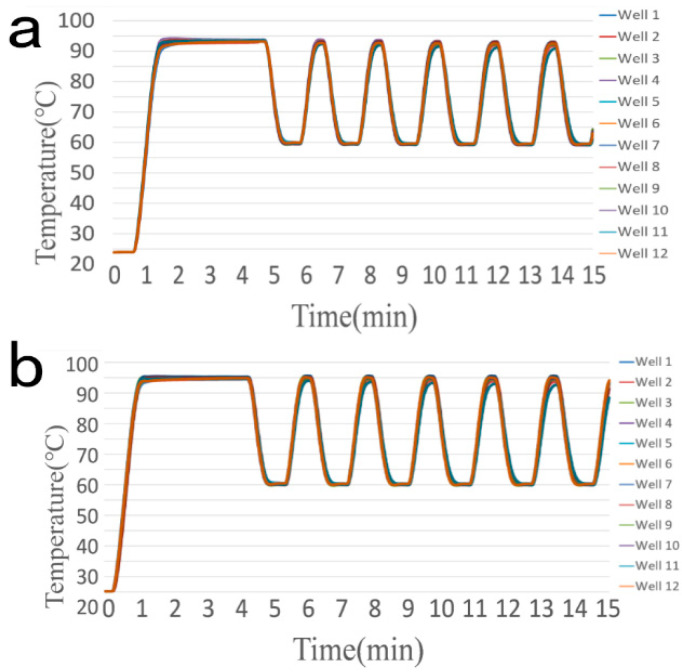
Temperature Curve of the Commercial PCR Amplification Program. (**a**) Before upgrade (the pre-denaturation step at 95 °C for 3 min, the denaturation step at 95 °C for 15 s, and the annealing/extension step at 60 °C for 35 s); (**b**) After the upgrade (the pre-denaturation step at 96.8 °C for 3 min, the denaturation step at 97.2 °C for 15 s, and the annealing/extension step at 60.5 °C for 35 s).

**Figure 3 bioengineering-09-00237-f003:**
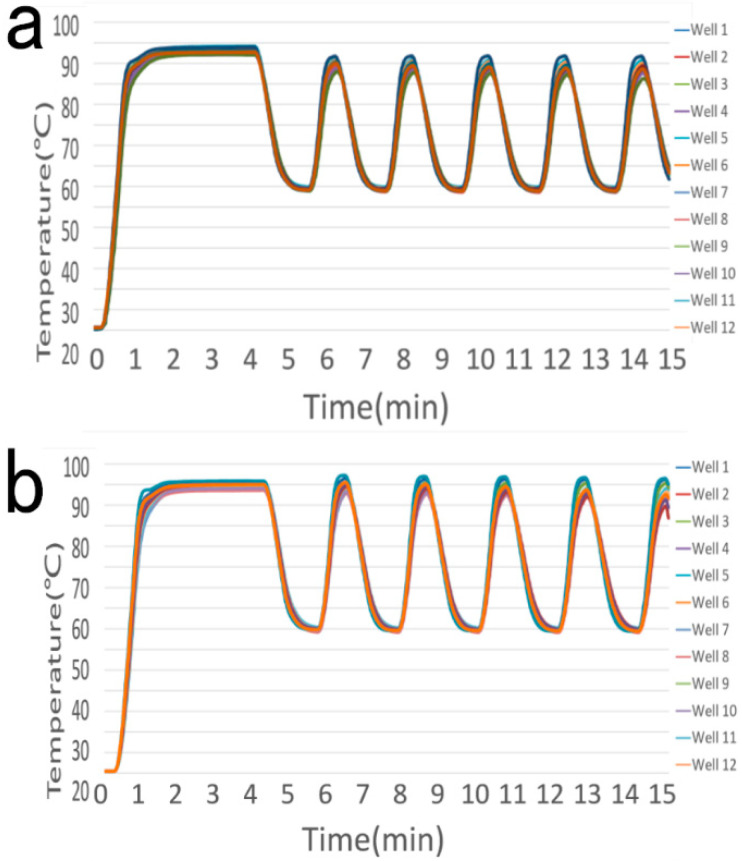
Temperature Curve of the Homemade PCR Amplification Program. (**a**) Before the upgrade (the pre-denaturation step at 95 °C for 3 min, the denaturation step at 95 °C for 15 s, and the annealing/extension step at 60 °C for 35 s); (**b**) After the upgrade (the pre-denaturation step at 97.2 °C for 3 min, the denaturation step at 100.4 °C for 15 s, and the annealing/extension step at 60.6 °C for 35 s).

**Figure 4 bioengineering-09-00237-f004:**
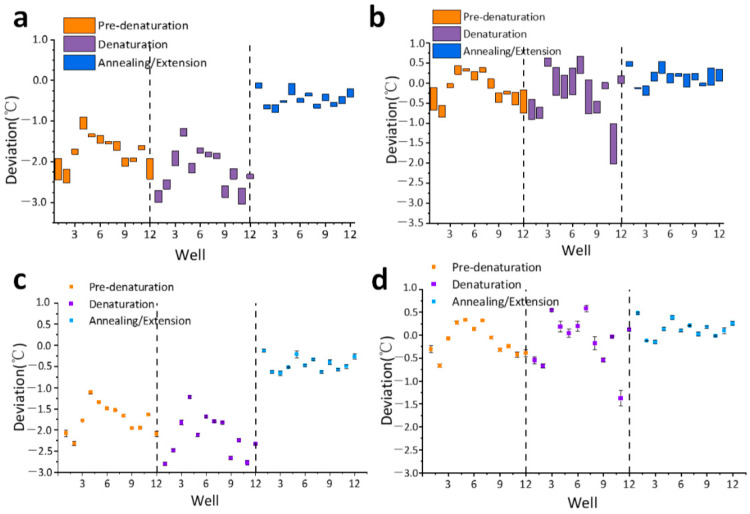
The actual temperature sampling of the reagent in the three steps of temperature cycle with 12 wells in the commercial PCR instrument. (**a**) The deviation range obtained from sampling in the initial temperature cycle; (**b**) The deviation range obtained from sampling in the upgraded temperature cycle; (**c**) The average value and standard deviation of sampling in the initial temperature cycle; (**d**) The average value and standard deviation of sampling in the upgraded temperature cycle.

**Figure 5 bioengineering-09-00237-f005:**
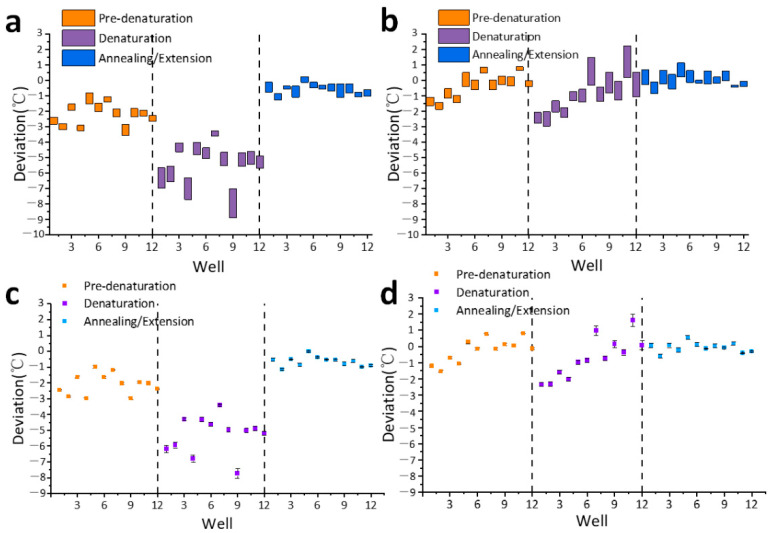
The actual temperature sampling of the reagent in the three steps of temperature cycle with 12 wells in the commercial PCR instrument. (**a**) The deviation range obtained from sampling in the initial temperature cycle; (**b**) The deviation range obtained from sampling in the upgraded temperature cycle; (**c**) The average value and standard deviation of sampling in the initial temperature cycle; (**d**) The average value and standard deviation of sampling in the upgraded temperature cycle.

**Figure 6 bioengineering-09-00237-f006:**
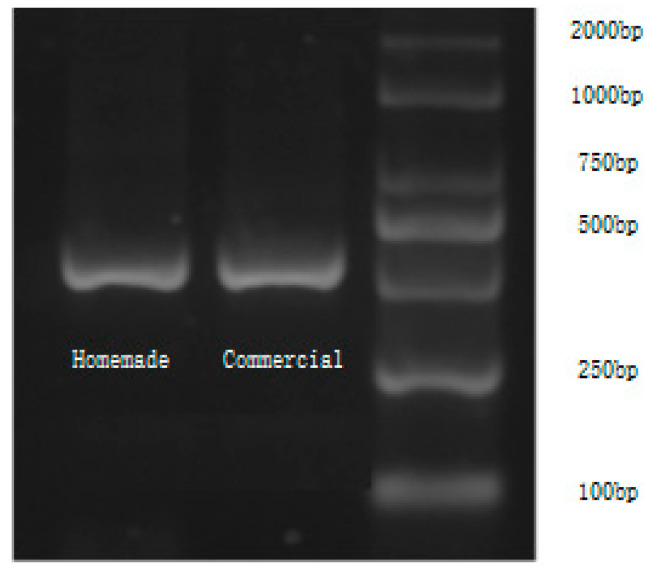
Results of DNA amplification of Escherichia coli using commercial PCR (LifeECO) and Homemade PCR with upgraded temperature cycle. The image shows the relative intensity of targeted genes amplified by the commercial PCR and homemade PCR and analyzed using ImageJ software.

**Figure 7 bioengineering-09-00237-f007:**
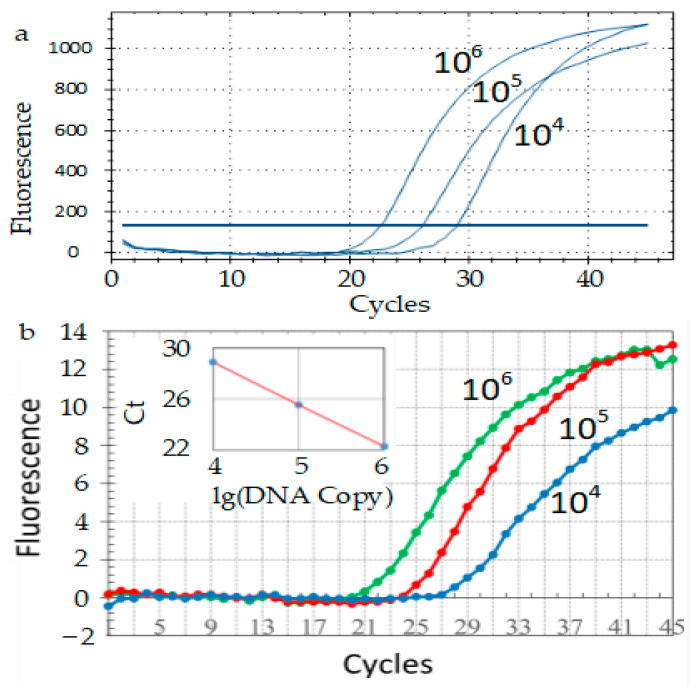
Results of DNA amplification of SARS-CoV-2 CDC positive plasmid. (**a**) Commercial QPCR instrument (Bio-Rad CFX Connect, Bio-Rad, CA, USA); (**b**) Homemade PCR with upgraded temperature cycle.

**Table 1 bioengineering-09-00237-t001:** The primer sequences.

Target	Primer Sequences
SARS-CoV-2	F: GGG GAA CTT CTC CTG CTA GAA T
	R: CAG ACA TTT TGC TCT CAA GCT G
Escherichia Coli	F: AGA GTT TGA TCC TGG CTC AG
	R: GWA TTA CCG CGG CKG CTG

## Data Availability

No conflict of interest exits in the submission of this manuscript. I would like to declare on behalf of my co-authors that the work described was original research that has not been published previously, and not under consideration for publication elsewhere, in whole or in part.

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
