# Peer review of "An Optimized Thermal Feedback Methodology for Accurate Temperature Control and High Amplification Efficiency during Fluorescent qPCR"

_bioengineering, 2022, doi:10.3390/bioengineering9060237_

Round 1
Reviewer 1 Report
In this manuscript, Wang et al, monitored the actual temperature derivations of wells in a home-made thermocycler. Based on that, the authors proposed an upgraded temperature cycle, showing smaller deviation and stable temperature control. Finally, as practical examples, they achieved efficient PCR amplification and qPCR detection, comparable to the commercial machines.
This study is a good fit for Bioengineering. Probably the engineers from thermocycler manufacturing field are also interested in this research. The measurement, result and conclusion were cleared described, with only a few edits recommended before publication. All my concerns are minor.
- Measurement strategy
The authors used a PT1000 thermistors, basically a single temperature sensor in a tube. Can they keep the PCR tube well capped during the reaction? Can the thermocycler lib be closed well?
For commercial thermocyclers, there is a self-diagnosis protocol. To check the machine, people usually use the temperature verification kit, which will give much more accurate reading and systematic analysis. These kits are commonly available. Can the author use this kit to monitor the temperature and verify their updated program?
- Discussion about future applications
The applications may be limited because we cannot ask new users to change their temperature setting all the times in order to achieve the better temperature control in the tubes. Instead, it is more about manufacturer’s responsibility to make sure the machine itself is reliable and people usually prefer qualified and verified thermocycler, even though it may be more expensive. Have the author thought about how to improve their machine, instead of simply updated program. This part may be beyond the scope of this manuscript, but definitely worth a discussion as potential directions.
- Diagram of PCR machine
The authors described the component or module of their homemade real-time PCR system. It is informative but not very intuitive. Can they draw a figure or diagram to show their design? Or a labelled picture is helpful, but much less elegant.
- Figure 2 and 3 cycle
In Figure 3, the x-axis is time (min), can they also label the cycle number and the PCR stage? Can they move the section of “well 1, well 2….” to the right side so people can align the x-axis value with the curve much more easily? Besides, how many cycles they have tested? The full image probably worth a supplementary figure.
- Figure 4 and 5.
It will be better if the author can draw a dashed line to separate these stages because these colors are difficult to distinguish.
- Figure 6.
The result is clear but not the gel. First, can they reverse the color, to show the clear background and dark band of DNA? People usually show the PCR with 2000bp ladder on the top, i.e. DNA is running from top to bottom.
What is the meaning of the x-axis “distance”? I guess it is just the pixel of the DNA image. The author measured the density using ImageJ and already clearly show the data of 92.5% (no replicates) in the main paragraph. So I think the bottom part of this figure can be simply removed. The gel image itself is sufficient.
Compared to the image, it is much more important to include more replicates, to prove the homemade machine achieved comparable efficiency to the commercial one. Then the authors can prepare a column diagram to show the quantitative measurement.
Author Response
1、Measurement strategy
The authors used a PT1000 thermistors, basically a single temperature sensor in a tube. Can they keep the PCR tube well capped during the reaction? Can the thermocycler lib be closed well?
For commercial thermocyclers, there is a self-diagnosis protocol. To check the machine, people usually use the temperature verification kit, which will give much more accurate reading and systematic analysis. These kits are commonly available. Can the author use this kit to monitor the temperature and verify their updated program?
Thank you for your suggestions. We measured the temperature in a 0.2 ml PCR tube containing 40 µ l of mineral oil (m8410, sigma Aldrich, Mo, USA). Use PT1000 thermistor (Japan lindenko) and temperature controller to record data. The specific operation method is to open two holes on the PCR tube cover that only pass through the sensing line of PT1000 thermistor, and the diameter of the hole is 0.85mm. The sensing line is perfectly combined with the aperture. At the same time, we use the bonding method to block the gap between the line and the aperture to keep warm. When the temperature sensor is buried in mineral oil, the mineral oil will also seal the temperature probe and play the role of thermal insulation. We use a temperature measuring device with combined PCR tube mode, which can be randomly combined into the temperature detection module of various commercial PCR equipment. The kit can be used to monitor temperatures and verify them.
2、Discussion about future applications
The applications may be limited because we cannot ask new users to change their temperature setting all the times in order to achieve the better temperature control in the tubes. Instead, it is more about manufacturer’s responsibility to make sure the machine itself is reliable and people usually prefer qualified and verified thermocycler, even though it may be more expensive. Have the author thought about how to improve their machine, instead of simply updated program. This part may be beyond the scope of this manuscript, but definitely worth a discussion as potential directions.
Thank you for your suggestions on the paper. We are considering further optimization of the temperature distribution uniformity of the thermal cycle unit. This is our next research direction.
3、Diagram of PCR machine
The authors described the component or module of their homemade real-time PCR system. It is informative but not very intuitive. Can they draw a figure or diagram to show their design? Or a labelled picture is helpful, but much less elegant.
Thank you for your suggestions on the paper. We added our designed detection module to Fig.1
4、Figure 2 and 3 cycle
In Figure 3, the x-axis is time (min), can they also label the cycle number and the PCR stage? Can they move the section of “well 1, well 2….” to the right side so people can align the x-axis value with the curve much more easily? Besides, how many cycles they have tested? The full image probably worth a supplementary figure.
Thank you for your suggestions on this paper. The x-axis represents the temperature measurement time. In this experiment, the pre denaturation step at 95 ° C for 3 min, the denaturation step at 95 ° C for 15 s, and the annual-ing / extension step at 60 ° C for 35 s are set. We counted the temperature detection for 15 minutes, including three temperature regions from room temperature to pre denaturation to denaturation to annealing / extension. At the same time, five PCR cycles were detected for experimental verification.
5、Figure 4 and 5.
It will be better if the author can draw a dashed line to separate these stages because these colors are difficult to distinguish.
Thank you for your suggestions on this paper. We have made changes in the paper.
6、Figure 6.
The result is clear but not the gel. First, can they reverse the color, to show the clear background and dark band of DNA? People usually show the PCR with 2000bp ladder on the top, i.e. DNA is running from top to bottom.
What is the meaning of the x-axis “distance”? I guess it is just the pixel of the DNA image. The author measured the density using ImageJ and already clearly show the data of 92.5% (no replicates) in the main paragraph. So I think the bottom part of this figure can be simply removed. The gel image itself is sufficient.
Compared to the image, it is much more important to include more replicates, to prove the homemade machine achieved comparable efficiency to the commercial one. Then the authors can prepare a column diagram to show the quantitative measurement.
Thank you for your suggestions on this paper. The picture has been changed in the body. For the time being, we only used E. coli for experimental verification, and later we used more samples for demonstration.
Reviewer 2 Report
This paper describes a new approach of performing real-time PCR by directly measuring the reaction temperature in the reaction reagent. In this way, the actual temperature of the reaction mixture is closer to the setting temperature. The new approach of temperature measurement is validated by improving the temperature control system of a commercial real-time PCR system and a home-built system. The new approach is also validated by comparing the performance of a home-built real-time PCR system with a Bio-rad commercial real-time PCR systems.
I have a few comments for the authors:
- It is not clear how the temperature of the reaction mixture was measured in the work. A scheme and a photograph of the home-built system will be helpful to better understand the work.
- The comparison of the home-built system with commercial system did not show obvious improvement in the quantitative PCR result although the temperature measurement was supposed to be more accurate. It would be helpful if the comparison of quantitative PCR result of the same home-built system with traditional and new temperature approaches is provided.
- There are a few minor writing mistakes, e.g., line 91-92.
Author Response
1.It is not clear how the temperature of the reaction mixture was measured in the work. A scheme and a photograph of the home-built system will be helpful to better understand the work.
Thank you for your suggestions. We measured the temperature in a 0.2 ml PCR tube containing 40 µ l of mineral oil (m8410, sigma Aldrich, Mo, USA). Use PT1000 thermistor (Japan lindenko) and temperature controller to record data. The specific operation method is to open two holes on the PCR tube cover that only pass through the sensing line of PT1000 thermistor, and the diameter of the hole is 0.85mm. The sensing line is perfectly combined with the aperture. At the same time, we use the bonding method to block the gap between the line and the aperture to keep warm. When the temperature sensor is buried in mineral oil, the mineral oil will also seal the temperature probe and play the role of thermal insulation. We added the working principle diagram in picture 1.
2.The comparison of the home-built system with commercial system did not show obvious improvement in the quantitative PCR result although the temperature measurement was supposed to be more accurate. It would be helpful if the comparison of quantitative PCR result of the same home-built system with traditional and new temperature approaches is provided.
Thank you for your suggestions on this paper. We used a homemade PCR instrument (without fluorescence detection system) and a commercial PCR instrument to amplify E. coli DNA. Products obtained from commercial PCR instrument (lifeeco) and self-made PCR instrument (492bp). The gel strength was calculated by ImageJ, and the amplification efficiency was about 92.5% of that of commercial PCR. The strength and position of the bands in the electrophoresis map showed that both commercial PCR and self-made PCR could efficiently amplify the correct products of E. coli. At the same time, the amplification efficiency of self-made PCR instrument is very close to that of commercial PCR instrument. We have temporarily conducted electrophoresis experiments to demonstrate the experimental results. In the future, we will use real-time fluorescence method for experimental demonstration.
3.There are a few minor writing mistakes, e.g., line 91-92.
Thank you for your suggestions. We have revised them in the paper.